# Characterization of a New Temperate *Escherichia coli* Phage vB_EcoP_ZX5 and Its Regulatory Protein

**DOI:** 10.3390/pathogens11121445

**Published:** 2022-11-30

**Authors:** Ping Li, Shanghai Yong, Xin Zhou, Jiayin Shen

**Affiliations:** 1Institute of Comparative Medicine, College of Veterinary Medicine, Yangzhou University, Yangzhou 225009, China; 2Jiangsu Co-Innovation Center for Prevention and Control of Important Animal Infectious Diseases and Zoonoses, Yangzhou University, Yangzhou 225009, China; 3Joint International Research Laboratory of Agriculture and Agri-Product Safety, The Ministry of Education of China, Yangzhou University, Yangzhou 225009, China; 4The Third People’s Hospital of Shenzhen, Shenzhen 518112, China

**Keywords:** temperate phage, biological characteristics, genomic analysis, lytic–lysogenic, virulence

## Abstract

The study of the interaction between temperate phages and bacteria is vital to understand their role in the development of human diseases. In this study, a novel temperate *Escherichia coli* phage, vB_EcoP_ZX5, with a genome size of 39,565 bp, was isolated from human fecal samples. It has a short tail and belongs to the genus *Uetakevirus* and the family *Podoviridae*. Phage vB_EcoP_ZX5 encodes three lysogeny-related proteins (ORF12, ORF21, and ORF4) and can be integrated into the 3′-end of *guaA* of its host *E. coli* YO1 for stable transmission to offspring bacteria. Phage vB_EcoP_ZX5 in lysogenized *E. coli* YO1+ was induced spontaneously, with a free phage titer of 10^7^ PFU/mL. The integration of vB_EcoP_ZX5 had no significant effect on growth, biofilm, environmental stress response, antibiotic sensitivity, adherence to HeLa cells, and virulence of *E. coli* YO1. The ORF4 anti-repressor, ORF12 integrase, and ORF21 repressors that affect the lytic–lysogenic cycle of vB_EcoP_ZX5 were verified by protein overexpression. We could tell from changes of the number of total phages and the transcription level of phage genes that repressor protein is the key determinant of lytic-to-lysogenic conversion, and anti-repressor protein promotes the conversion from lysogenic cycle to lytic cycle.

## 1. Introduction

Phages are viruses that infect bacterial cells, and 1 × 10^31^ phages are estimated to be found on earth. These viruses are extremely abundant and ubiquitous in nature and can be found in aquatic environments, soil, and human/animal ecosystems [1,2,3,4]. Phages can be sorted into obligately lytic and temperate phages according to their life cycles. Similar to other viruses, phages can highjack host cells to replicate. Obligately lytic phages release their assembled phage progeny into the environment via host cell lysis [5]. Temperate phages can integrate their DNA into host chromosomes or exist as plasmids and, in most cases, have no lytic effect on their hosts [6]. The increasingly prevalent antibiotic resistance of bacteria has resulted in phages receiving increased attention as a substitute for traditional antibiotics. Phages have the advantages of strong specificity, fewer adverse effects, and relatively low drug development costs [7,8,9]. Obligately lytic phages have been widely used to inactivate and control pathogens in the food, agriculture, veterinary, clinical, and industrial sectors, among other fields. Temperate phages, such as lambda phages, have both obligately lytic and lysogenic cycles [10]. Most temperate phages encode transposases, integrases, virulence factors, and other potential safety hazards, resulting in limited applications [11,12,13]. Under most conditions, phage integration in lysogenic cells is stable enough to maintain its existence even after multiple cell divisions. Only a small number of phages can be induced spontaneously, and the induction of most prophages requires certain conditions, such as ultraviolet radiation, high osmotic pressure, H_2_O_2_, and mitomycin C [14,15,16,17]. Studies have shown that the integration of prophages increases growth, antibiotic resistance, biofilm formation, adhesion, motility, environmental stress response, and virulence of their hosts [18,19,20,21]. However, some phage genes may have opposite effects on the growth of host cells since the expression of phage proteins affects the availability of host cell resources [22].

Phages are the main components (>97%) of human enteroviruses. It is estimated that there are approximately 10^8^–10^9^ phage particles per gram of human feces and 10^15^ particles in the gastrointestinal tract [23,24]. There is considerable diversity among phages, and phages in the human intestine are mainly non-enveloped dsDNA *Caudovirales*, ssDNA *Microviridae* and ssDNA filamentous phages [25,26]. Metagenomic studies have shown that most intestinal bacteria carry prophage genes [25,27]. By infecting specific populations of bacteria, phages can alter the structures of microbiota or their phenotypes, leading to intestinal homeostasis or dysbiosis and even chronic infectious and autoimmune diseases [28,29]. Temperate phages can strongly promote the virulence and adaptability of bacterial hosts by encoding virulence genes and then play a pro-inflammatory role in the intestines [4,30]. Temperate phages can increase the adaptability of symbiotic bacteria by eliminating repeated infections or lysogenic transformations to help maintain a healthy intestinal environment [31]. The induction of temperate phages leads to the death of bacterial hosts and changes population dynamics and intestinal ecological balance [32]. Temperate phages play an important role in research because of their fundamental role in intestinal health.

Research on intestinal phages has focused chiefly on metagenomics, with few studies reporting on a single individual. In this study, a novel temperate *Escherichia coli* phage from the gut was isolated. The biological and genomic characteristics, life cycles, and host virulence of the isolated phages were analyzed to gain valuable insights into intestinal temperate phages.

## 2. Materials and Methods

### 2.1. Bacteria and Culture Conditions

*E. coli* YO1 was isolated from fecal samples at Yangzhou University in 2021 and cultured in Luria-Bertani (LB) broth at 37 °C and 220 rpm. The 16S rRNA genes were amplified using the primers 27F and 1492R for identification. *dinB*, *icdA*, *pabB*, *polB*, *putP*, *trpA*, *trpB*, and *uidA* were used for multi-locus sequence typing (MLST) [33]. O antigen was determined by Gao Song’s team from Veterinary College of Yangzhou University using agglutination tests [34]. H antigen was determined by PCR and sequencing of *fliC* using primer fliCF/fliCR [35]. K1 antigen was determined by PCR and sequencing of *kpsM* using primer K1F/K1R [36]. The important virulence genes *aatA*, *aggR*, *eae*, *bfpA*, *ltA*, *stA*, *ipaH*, *stx1*, and *stx2* of enteropathogenic *E. coli* (EPEC), enteroaggregative *E. coli* (EAEC), Shiga toxin-producing *E. coli* (STEC), enterotoxigenic *E. coli* (ETEC), and enteroinvasive *E. coli* (EIEC) were identified by PCR [37]. Antibiotic resistance was determined using standard Kirby-Bauer disk diffusion. Data analyses were performed according to the guidelines of the standardized protocol of the National Committee for Clinical Laboratory Standards and designated as R (resistant), I (intermediate sensitive), and S (sensitive). The primers used in this study are listed in Appendix A. 

### 2.2. Phage Isolation and Purification

Phage vB_EcoP_ZX5 was isolated from the fecal samples at Yangzhou University in 2021. Briefly, 1 g of fecal sample was completely mixed with 1 mL of PBS and centrifuged at 8000× *g* for 5 min. The resulting supernatants were passed through a 0.22 μm filter. Then, the supernatant filtrates were added to 5 mL of logarithmic phase (OD_600_ = 0.6) *E. coli* YO1 and cultured at 37 °C for 2–4 h. The lysate was centrifuged at 12,000× *g* for 2 min, and the phages in the supernatants were purified using the double plate method. Briefly, 300 μL of *E. coli* YO1 and 100 μL of a ten-fold dilution series of phages were added to 5 mL of top LB soft agar and then poured onto an LB agar plate and cultured at 37 °C overnight. The next day, a single plaque was picked from the plate, inoculated into 5 mL of logarithmic phase host, and cultured at 37 °C for 2–4 h. The cultures were centrifuged at 12,000× *g* for 2 min, and the lysates were passed through a 0.22-μm filter. These steps were repeated at least three times until a single phage was obtained. 

PEG/NaCl was used to enrich phage particles. The phages were added to 100 mL of logarithmic phase host and cultured at 37 °C and 220 rpm for 2–4 h. The cultures were centrifuged at 8000× *g* for 10 min. Then, DNase I and RNase A (1 µg/mL) were added to the lysate and incubated at 37 °C for 30 min. Next, 1 M NaCl was added to the supernatant, which was then placed in an ice bath for 1 h. After centrifugation at 8000× *g* for 10 min, 10% (*w*/*v*) polyethylene glycol 8000 (PEG 8000) was added to the lysate and precipitated overnight at 4 °C. After centrifugation at 8000× *g* for 20 min, the precipitate was resuspended in PBS. The phage suspension was stored at 4 °C.

### 2.3. DNA Isolation and Sequencing 

Phage vB_EcoP_ZX5 DNA was extracted using the Virus Genomic DNA/RNA Extraction Kit (Tiangen, Beijing, China). Whole-genome sequencing of the phage was performed by Shanghai Bioengineering Co. Ltd. Phage DNA fragments with a length of approximately 500 bp were randomly interrupted by a Covaris ultrasonic crusher (Covaris, Woburn, MA, USA) and then purified using Hieff NGS™ DNA selection beads (Yeasen, Shanghai, China). The sequencing library was constructed using the NEB Next^®^ Ultra™ DNA Library Prep Kit for Illumina^®^ (NEB, Ipswich, MA, USA), including terminal repair, adaptor ligation, DNA purification, and library amplification. After passing the quality test, the DNA library was sequenced using an Illumina hiseqpe150 sequencing platform. The original sequencing data were filtered at first and then were assembled using New Blew 3.0.

### 2.4. Genome Analysis

The tRNAs in the genome were predicted using tRNAscan-SE 2.0 (http://lowelab.ucsc.edu/tRNAscan-SE/, accessed on 6 April 2022) [38]. The coding domain sequences (CDSs) were predicted using the RAST annotation server web (https://rast.nmpdr.org/, accessed on 10 March 2022) and aligned using BLASTX in NCBI (https://blast.ncbi.nlm.nih.gov, accessed on 6 April 2022) [39]. Virulence and drug resistance genes were compared using VFPB (http://www.mgc.ac.cn/VFs/main.htm, accessed on 6 April 2022) and CARD (https://card.mcmaster.ca, accessed on 6 April 2022) [40,41]. The genome was visualized using Easyfig 2.2.5 [42]. Genomic blast analysis was performed using the BLAST Ring Image Generator (BRIG) [43]. Multiple sequence alignment of the genomes was performed using Easyfig 2.2.5. Dot plots of the whole-genome analysis were generated using Gepard [44]. Phylogenetic tree analysis of the phage terminase large subunit was performed by MEGA 5.0 using the neighbor-joining method with 1000 bootstraps. The promoters of phage were predicted by Promoter Hunter (http://www.phisite.org/main/index.php?nav=tools&nav_sel=hunter, accessed on 10 May 2022).

### 2.5. Transmission Electron Microscopy (TEM)

The purified phage vB_EcoP_ZX5 was incubated on carbon grids (200 mesh) for 10 min, stained with 2% phosphotungstic acid for 1 min, and dried for 30 min. The morphology of the phage was examined using transmission electron microscopy (TEM) (HT7800; Hitachi, Tokyo, Japan).

### 2.6. Phage Adsorption Test

The phage vB_EcoP_ZX5 was added to a logarithmic phase *E. coli* YO1 (1 × 10^8^ CFU/mL) at an multiplicity of infection (MOI) of 0.1 and cultured at 37 °C [45]. The phage titer in the culture supernatant was measured every 5 min. The experiment was repeated in triplicate. The adsorption time and efficiency of the phages were then calculated. 

### 2.7. One-Step Growth Curve

The phage vB_EcoP_ZX5 was added to a logarithmic phase *E. coli* YO1 (1 × 10^8^ CFU/mL) at an MOI of 0.1. The mixture was incubated at 37 °C for 10 min and then centrifuged at 12,000× *g* for 1 min to remove free phages. The precipitate was resuspended in LB broth (time zero) and cultured at 37 °C and 220 rpm. Three parallel samples were collected every 10 min and centrifuged at 12,000× *g* for 1 min. Phage titer in the supernatant was determined using the double agar plate method. Burst size was calculated as follows: burst size = number of released phages/number of infected bacteria. The experiments were repeated in triplicate.

### 2.8. Phage Stability Studies

#### 2.8.1. Thermal Stability Test

To test for thermal stability, 300 µL of phage suspension (10^8^ PFU/mL) was incubated at 4 °C, 40 °C, 50 °C, 60 °C, and 70 °C for 20, 40, and 60 min. The resulting phage titers were determined using the double-layer plate method. The experiments were repeated in triplicate.

#### 2.8.2. pH Stability Test

To test for pH stability, 300 µL of phage suspension (10^8^ PFU/mL) was incubated at different pH buffers (pH 1–13) and then incubated at 37 °C for 20, 40, and 60 min. The resulting phage titers were determined using the double-layer plate method. The experiments were repeated in triplicate.

### 2.9. Screening and Subculture of Phage Lysogenic Bacteria

Phage vB_EcoP_ZX5 was added to the logarithmic phase *E. coli* YO1 with an MOI of 0.1 and co-cultured in LB broth at 37 °C for 12 h. The cultures were then scribed to obtain monoclonal bacteria. The monoclonal strains were randomly selected and cultured in LB broth at 37 °C for 12 h. Next, the lysogenized bacteria were verified by PCR amplification of phage integrase (1F/1R) and terminase large subunit (2F/2R). The lysogenized bacterial cultures were subcultured again. These steps were repeated to obtain lysogenic *E. coli* YO1+ after ten passages. Primer 3F and 3R are from the 3′-end of *guaA* in *E. coli* YO1, and primer 3F2 and 3R2 are from the 5′-end of *integrase* in phage vB_EcoP_ZX5. Primer 3F/3R was used to amplify the bacterial attachment site (attB) in *E. coli* YO1, resulting in a 286bp DNA1 fragment. Primers 3F/3R2 and 3F2/3R were used to amplify the phage integration site (attP) attL and attR in *E. coli* YO1+, respectively, resulting in a 214bp DNA2 fragment and a 281bp DNA3 fragment. The sequences of DNA1, DNA2 and DNA3 were verified by Sanger sequencing. The titer of free phages in the lysogenized bacterial culture supernatant was determined using a double-layer plate method. 

### 2.10. Phage Infection Characteristics

Different doses of phages (MOI = 0.1, 1 and 10) were added to the logarithmic phase host (10^8^ CFU/mL) and cultured in LB broth at 37 °C and 220 rpm. The number of phages was determined every 30 min. The titer of total phages and prophages were determined by qPCR using primers 4F/4R and 3F2/3R. The titer of free phages in the supernatant was determined using a double-layer plate method. Primer 4F/4R was used to amplify the sequence of terminase large subunit in phage vB_EcoP_ZX5, which can amplify 230 bp fragments in all phage genomes including lytic and lysogenic cycle. Because primers 3F2 and 3R are derived from phage and bacteria, respectively, 3F2/3R can only amplify 281 bp fragments in lysogenic strains integrated with phage vB_EcoP_ZX5 in lysogenic cycle. The experiment was repeated three times. qPCR reactions were performed using a Roche Light Cycler 480 II PCR instrument (Roche, Basel, Switzerland) and ChamQ Universal SYBR qPCR Master Mix (Vazyme, Nanjing, China), according to the manufacturer’s instructions. The cycling conditions were a step at 95 °C for 30 s, followed by 40 cycles of 10 s at 95 °C and 30 s at 60 °C, and finally 95 °C for 15 s, 60 °C for 60 s, 95 °C for 15 s, and 37 °C for 2 min. 

### 2.11. Construction of Overexpression Plasmids

The candidate genes that affected the phage vB_EcoP_ZX5 lytic–lysogenic cycles were verified by gene overexpression. Plasmid pGEX-6P-1 (GenBank: U78872.1) carrying GST tag was used to express phage genes. *Gp4 anti-repressor*, *gp12 integrase*, and *gp21 repressor* with his tag were amplified using primers 5F/5R to 7F/7R and cloned into the plasmid pGEX-6P-1 using the ClonExpress II One Step Cloning Kit (Vazyme). The products were transformed into DH5α competent cells, and the monoclonal strains were cultured for sequencing identification (8F/8R). The positive plasmid pGEX-6P-x was extracted using Plasmid Mini Kit (Vazyme) and stored at −20 °C. *E. coli* YO1 was cultured in the logarithmic phase, and competent cells were prepared. The plasmids were electro-transformed into *E. coli* YO1 using an electroporator (Eppendorf, Hamburg, Germany). *E. coli* YO1 transformed with plasmid pGEX-6P-x was identified by PCR (8F/8R) and was cultured to logarithmic phase in LB broth at 37 °C. *E. coli* YO1 carrying the empty plasmid pGEX-6P-1 containing GST tag (GST) was used as a negative control. Isopropyl ß-D-1-thiogalactopyranoside (IPTG) at a final concentration of 1 nM was added to the bacterial culture and then grown for 6 h. The protein expression of plasmid pGEX-6P-x was characterized by western blotting using anti-His-Tag mouse monoclonal antibody (Cwbio, Taizhou, China) and goat anti-mouse IgG conjugated to horse-radish peroxidase (Cwbio). 

### 2.12. Effect of Gene Overexpression on Phage Cycles

*E. coli* YO1 with plasmid pGEX-6P-x was cultured to logarithmic phase in LB broth at 37 °C. IPTG at a final concentration of 1 nM and phage vB_EcoP_ZX5 at a final concentration of 10^7^ PFU/mL were added to the bacterial culture and then grown for 6 h. After adding phages, 3 mL of cultures were collected every hour. Next, 1 mL of culture was rapidly frozen and stored at –80 °C to determine the titer of total phages and prophages using qPCR. Then, 1 mL of culture was centrifuged at 12,000× *g* for 2 min, and the titer of free phages in the supernatants was determined using the double-layer plate method. Lastly, 1 mL of culture was centrifuged at 12,000× *g* for 2 min, and the bacterial precipitate was stored at –80 °C until the transcription level of phage genes was analyzed by qPCR. The experiment was repeated three times.

The total RNA of the bacteria was extracted by TRNzol (Tiangen, Beijing, China) as follows: the bacterial precipitation was resuspended with 200 μL of TRNzol, 200 μL of chloroform was added and centrifuged at 12,000× *g* for 15 min, the supernatant was transferred to a new centrifuge tube with 200 μL of isopropanol, the tube was placed in an ice bath for 1 h and then centrifuged at 12,000× *g* for 15 min, the resulting supernatant was discarded, and the RNA was dissolved in RNase-free water. RNA samples were reverse-transcribed using HiScript^®^ III RT SuperMix for qPCR (Vazyme). The transcription levels of phage integrase (9F/9R), repressor (10F/10R), anti-repressor (11F/11R), lysin (12F/12R) and holin (13F/13R) were measured by qPCR. The experiment was repeated three times. The cycling conditions were a step at 95 °C for 30 s, followed by 35 cycles of 10 s at 95 °C and 30 s at 60 °C, and finally 95 °C for 15 s, 60 °C for 60 s, 95 °C for 15 s, and 37 °C for 2 min.

### 2.13. Bacterial Growth Curve and Resistance to Environmental Stress

Overnight cultured *E. coli* YO1 and *E. coli* YO1+ were inoculated into a fresh medium (10^7^ CFU/mL), and the OD_600_ of bacteria was measured using a Nanophotometer Ultramicro Spectrophotometer (Implen, Munich, Germany) every 30 min. The experiment was repeated three times.

The responses of the *E. coli* YO1 and *E. coli* YO1+ to external stress were evaluated by measuring the change in the bacterial count under different stresses. The logarithmic phase bacteria (5 × 10^7^ CFU/mL) were treated with acid buffer (pH 3), alkaline buffer (pH 10), and 30 mM H_2_O_2_ at 37 °C for 10 min. The logarithmic phase bacteria were treated at 50 °C for 10 min to test for heat stress. After treatment, cell viability was measured using the plate counting method. The experiment was repeated three times.

### 2.14. Biofilm Assay

The overnight cultured *E. coli* YO1 (group YO1) and *E. coli* YO1+ (group YO1+) were transferred to fresh medium in 96-well polystyrene plates (Corning Costar, Corning, NY, USA) at a ratio of 1:100, setting PBS as blank control. The bacteria of each group were cultured at 37 °C, in which the culture medium was replaced with fresh LB every 24 h. After three days of culture, biofilm formation was measured using crystal violet staining. Each well was washed thrice with ddH_2_O and dried. Crystal violet (200 μL) was added to each well and incubated for 10 min. This was followed by washing each well thrice with ddH_2_O three times and drying. Glacial acetic acid (200 μL) was added to each well and incubated at for 10 min. The OD_600_ of each well was measured using a smart microplate reader (Infinite M200 Pro; Tecan, Männedorf, Switzerland). The experiment was repeated three times.

### 2.15. Adhesion and Invasion Assays

HeLa cells were used to analyze the adhesion and invasion abilities of *E. coli* YO1 or *E. coli* YO1+ [46]. In short, HeLa cells were inoculated on a 24-well plate and were cultured in a 5% (*v*/*v*) CO_2_ incubator at 37 °C to form monolayer cells (10^5^ cells per well). The logarithmic phase *E. coli* YO1 or *E. coli* YO1+ cells were resuspended in the FBS free DEME medium (10^7^ CFU/mL), and then co-incubated with HeLa cells at 37 °C for 2h. For adhesion test, the cells were washed three times with PBS and then lysed with 0.5% (*v*/*v*) Triton X-100. The number of bacteria was calculated on LB plates. For the invasion test, after the cells were co-incubated with bacteria, ampicillin antibiotics (100 μg/mL) were added and incubated at 37 °C for 2 h to kill the bacteria on the surfaces of cell. Cells were cleaned by PBS for three times and then lysed with 0.5% (*v*/*v*) Triton X-100. 

### 2.16. Analyses of Serum Sensitivity and Virulence of E. coli YO1

Female BABL/C mice, aged 6–8 weeks, were purchased from the Experimental Animal Center of Yangzhou University. All animal experiments were performed in strict accordance with the Regulations for the Administration of Affairs Concerning Experimental Animals, approved by the Animal Welfare and Research Ethics Committee of Yangzhou University.

The blood of healthy mice was collected and immediately stored at 37 °C for 2 h and then centrifuged at 2000× *g* for 5 min to collect serum. A total of 100 μL of *E. coli* YO1 and *E. coli* YO1+ (10^4^ CFU/mL) were mixed with 100 μL of mouse serum. After incubation at 37 °C for 1 h, the cell viability was determined. The experiment was repeated three times.

*E. coli* YO1 and *E. coli* YO1+ were cultured overnight and resuspended in PBS after centrifugation. The mice were divided into three groups, with six mice per group. Mice were intraperitoneally injected with *E. coli* YO1 (10^9^ CFU), *E. coli* YO1+ (10^9^ CFU), or PBS at a total volume of 100 μL. The survival of the mice was recorded every day. Three mice were euthanized two days after infection. The main organs were collected and homogenized by dissection, and the bacterial load was measured using the plate counting method.

## 3. Results and Discussion

### 3.1. Isolation of the Phage vB_EcoP_ZX5

*E. coli* YO1 was identified as serotype O87:H52:K1 and ST83 sequence type. No virulence genes of EPEC, EAEC, STEC, ETEC, and EIEC were detected in *E. coli* YO1. The phage vB_EcoP_ZX5 isolated from human fecal samples formed small translucent round plaques (1 mm) on the bacterial lawn (Figure 1a). According to TEM analysis, the phage vB_EcoP_ZX5 had a capsid with a diameter of 60 ± 3 nm and a short tail with a length of 15 nm (Figure 1b). Although studies have shown that most of the complete temperate phages exist in guts belong to *Sipoviridae*, the temperate phage vB_EcoP_ZX5 isolated in this study belongs to *Podoviruses* [25]. 

### 3.2. Biological Features of the Phage vB_EcoP_ZX5

The phage adsorption process showed that the adsorption time of the phage vB_EcoP_ZX5 took place within the first 25 min (Figure 2a). The proportion of phages adsorbed to the bacterial surface reached 98.6% at the 25th minute. The curve shows that the phage titer increased from the 30th minute, indicating the release of offspring phages. The one-step growth curve of the phage vB_EcoP_ZX5 showed that its latent period was approximately 20 min and its burst size was approximately 700 PFU/cell (Figure 2b). The burst size of the phage vB_EcoP_ZX5 is 3–7 times that of most phages [47,48,49]. We speculate that such a large burst size may be related to the high adsorption rate (3.9 × 10^−3^ PFU/cell/min/mL) of the phage vB_EcoP_ZX5. Another phage S0112 has a larger burst size of 1170 PFU/cell, which may be related to its long latent period [50].

Thermal and pH stability tests showed that the phage vB_EcoP_ZX5 maintained a broad stability range. The phage vB_EcoP_ZX5 showed high stability at 4–50 °C and pH = 5–9. The phage lost 87/99.93% activity after 20/60 min at 60 °C, and completely inactivated after 20 min at 70 °C (Figure 2c). Phage was slightly inactivated (30/17%) at pH = 4/10 for 60 min, most inactivated (99.8/98.3%) at pH = 2/11 for 40 min, and completely inactivated at pH = 1/13 for 20 min (Figure 2d). 

### 3.3. Genome Characteristics of the Phage vB_EcoP_ZX5

The genome size of the phage vB_EcoP_ZX5 (MW722083) is 39,565 bp. The genome of vB_EcoP_ZX5 showed a high GC content of 49.00%. One tRNA (complement 14482 ... 14558) was annotated in the vB_EcoP_ZX5 genome, and virulence or drug resistance genes were not detected. A total of 53 open reading frames (ORFs) were annotated in the phage vB_EcoP_ZX5 genome, of which 23 (43%) were functional proteins (Appendix A). The ORFs in the vB_EcoP_ZX5 genome used ATG (49 ORFs) and GTG (4 ORFs) as initiation codons. These ORFs were divided into five categories: hypothetical protein, nucleic acid metabolism, phage structure, phage assembly, and phage lysis (Figure 3a). The phage structure, lysis, and DNA packaging genes are transcribed in the same direction, whereas DNA metabolism and hypothetical protein genes are transcribed in different directions. The four predicted promoters are shown in Figure 3, in which promoter *P_L_* may start the lytic cycle and promoter *P_R_* may start the lysogenic cycle.

Three lysogenic-related proteins were annotated as ORF4 anti-repressor, ORF12 integrase, and ORF21 repressor. Integrases are necessary for phages to integrate into the host genome by mediating site-specific recombination between the attP and attB [51,52]. Integrases can be grouped into two major families, namely tyrosine and serine recombinases, based on their mode of catalysis. Lambda phage integrase is a typical tyrosine integrase, comprising an N-terminal Arm-DNA binding region, an intermediate core-binding region that binds to regulatory DNA sequences, and a C-terminal catalytic region [53]. ORF12 is annotated as a tyrosine integrase in the phage vB_EcoP_ZX5 with an Arm-DNA binding domain at its N-terminus and a P4-like phage integrase C-terminal catalytic domain at its C-terminus. The attP is often adjacent to the P4-like phage integrase gene, whereas the attB are typically situated near tRNA genes [54]. The core sequence of the attachment site attP (attgagtgggaatgatt) is located at 190 bp at the 5′-end of the integrase in phage vB_EcoP_ZX5, and attB (attgagtgggaatgatt) is located at the 3′-end of the *guaA* in *E. coli* YO1. *E. coli* YO1 + was obtained after phage vB_EcoP_ZX5 integrated its genome into the 3′-end of *guaA* in *E. coli* YO1 (Figure 3b).

The binding of repressor to operator site can suppress the lytic cycle, while anti-repressor will competitively bind with repressor to initiate lytic cycle [55,56,57]. ORF21 and ORF4 were annotated as repressor and anti-repressor in phage vB_EcoP_ZX5. The C-terminus of the ORF21 repressor has a conserved domain of the HTH-type transcriptional regulator TreR, which functions in DNA-binding. The N-terminus of the ORF4 anti-repressor has a conserved domain in the N-terminal structure of the bro family. The function of the bro-family proteins is not clear, and they may be DNA-binding proteins that affect host DNA replication or transcription. 

### 3.4. Phylogenetic Analysis and Comparative Genomic Analysis of the Phage vB_EcoP_ZX5

We confirmed the taxonomic location of the phages isolated in this study. First, a phylogenetic analysis based on the phage terminase large subunit was performed, which showed that the phage vB_EcoP_ZX5 belongs to the cluster of the genus *Uetakevirus* and the family *Podoviridae* (Figure 4a). The whole-genome sequence point map analysis of the same phage used in the phylogenetic analysis further confirmed the taxonomy of the phage vB_EcoP_ZX5 (Figure 4b).

BLASTN in NCBI was used to align the genome similarity of the phage vB_EcoP_ZX5 with the genome published in the Nt library. The phage vB_EcoP_ZX5 had a sequence similarity (50–80%) to phages of the genus *Uetakevirus*, such as *E. coli* O157 phage type 10 (KP869108), *E. coli* O157 phage type 9 (KP869107), *Enterobacteria* phage phiV10 (DQ126339), *Escherichia* phage phiv142-3 (MN187550), *Escherichia* phage TL-2011b (JQ011317), and *Enterobacter* phage Tyrion (KX231829)

The phage vB_EcoP_ZX5 was found to have high sequence similarity to *E. coli* O157 phage type 10 (coverage 77%, indentity 98.42%), *Enterobacteria* phage phiV10 (coverage 77%, indentity 98.42%), *Escherichia* phage phiv142-3 (coverage 80%, indentity 98.42%), and *Enterobacteria* phage epsilon15 (coverage 37%, indentity 82%). Multiple sequence alignments of the phage vB_EcoP_ZX5, phiV10, phiv142-3, epsilon15, and *E. coli* O157 phage type 10 are shown in Figure 5. These phages had similar genome sizes (39 kb), GC contents (49%), and gene structures. The integrase (ORF12) of the phage vB_EcoP_ZX5 had high (98%) homology with the integrase of phiV10 (ORF31), phiv142-3 (ORF53), epsilon15, and *E. coli* O157 phage type 10 (ORF11). The sequence attgagtgggaatgatt has been proved to be the attP of phage phiv10 and epsilon15, and it was proved to be the attP of phage vB_EcoP_ZX5 in this study [58,59]. The repressor (ORF21) of the phage vB_EcoP_ZX5 phiV10 (ORF40), phiv142-3 (ORF43), and *E. coli* O157 phage type 10 (ORF3) had the same sequence, and they had 42% homology with the repressor (ORF19) of the phage epsilon15. The anti-repressor (ORF4) of phage vB_EcoP_ZX5 had low (39%) homology with the anti-repressor of phiV10 (ORF23) and phiv142-3 (ORF8).

ORF27-31, ORF50-53, ORF1-2, and ORF7 of the phage vB_EcoP_ZX5 were significantly different from those of the other phages (Figure 5). ORF27-31 is mainly involved in nucleic acid metabolism. ORF50-53 and ORF1-2 were annotated as hypothetical proteins. ORF7 encodes a tail protein. Phage tail proteins were considered the main components involved in phage recognition and mediate primary attachment to host cells [60,61]. ORF7 of the phage vB_EcoP_ZX5 encodes three conserved domains: the N-terminal domain is phage T7 like tail fibrin, the middle is phage endosialidase-like protein, and the C-terminal is the intramolecular chaperone of endosialidase. An α-2,8-linked polysialic acid (polySia) capsule confers immune tolerance to neuroinvasive, pathogenic prokaryotes such as *E.coli* K1 and *Neisseria meningitidis* and supports host infection by means of molecular mimicry [62,63,64]. Endosialidases are phage tailspike proteins that specifically bind and degrade the α 2,8-linked polysialic acid capsules in *E. coli* K1 [65,66,67].

### 3.5. Stable Passage of Phage vB_EcoP_ZX5 in Lysogenic Strain

The *E. coli* YO1+ was subcultured to determine the genetic stability of the phage vB_EcoP_ZX5. The results showed that the phage vB_EcoP_ZX5 existed in the 10th generation of the *E. coli* YO1+ genome. After culturing for 12 h, the phage titer in the supernatant of all progeny lysogenic bacteria was approximately 10^7^ PFU/mL. This indicates that the phage vB_EcoP_ZX5 can exist stably in *E. coli* YO1+ and be induced spontaneously. Since the discovery of spontaneous induction in the 1950s, reports on spontaneous induction by phages have gradually increased [68]. The titers of spontaneously induced free phages ranged from 10^2^ to 10^9^ PFU/mL [69,70,71,72]. Generally, phage induction occurs by activating SOS responses caused by cell stress [73,74]. The titer of free phages vB_EcoP_ZX5 reached 10^9^ PFU after 12 h of induction of lysogenic bacteria with mitomycin C at a final concentration 0.5 μg/mL. Evidence shows that phage induction can increase bacterial virulence, increase biofilm formation, and contribute to gene transfer [75,76,77,78]. Overall, phage induction and its potential impact on host biology have not yet been fully demonstrated.

### 3.6. Infectious Characteristics of the Phage vB_EcoP_ZX5 from the Lytic to Lysogenic Phase

Different doses of phages (MOI = 0.1, 1, and 10) were used to infect the hosts in the logarithmic phase (10^8^ CFU/mL). The titer of total phages, prophages and free phages was measured every 0.5 h (Figure 6). After addition, phages immediately infect their hosts and continually release many free phages. The titer of total phages tended to stabilize after reaching 10^9^ PFU/mL. The titer of prophages increased gradually, until it reached a steady-state (10^9^ PFU/mL). With an increase in prophages, the titer of free phages gradually decreased to 10^7^ PFU/mL. The data indicate that the trends of total phages, free phages, and prophages were similar when added at MOI = 0.1, 1, or 10. In the early stage of co-culture, the titer of free phages increased rapidly until reaching the highest level. In the middle of co-culture, the titer of prophages increased and that of the free phages gradually decreased. In the later stage of co-culture, after each bacterium integrated a phage genome, a proportion of 1% free phages existed in the culture. 

Because the dominant phage population in phage culture was changed from free phages to prophages, we firmly believe that there is some modulation behind this transformation. For example, the transcription regulatory factors CI and Cro encoded by phage lambda initiate the lytic cycle of phage when the bacteria react with SOS [73,79]. VqmA, encoded by the *Vibrio cholerae* phage vp882, can be combined with 3,5-dimethylpyrazine-2-ol (DPO), which is produced by the host at a high cell density to initiate the phage lytic cycle [80]. Aimp, AimR, and AimX, encoded by the *Bacillus subtilis* phage phi3t, can rely on the small molecule communication by lysogenic bacteria to make the lysis–lysogeny decision, and high concentrations of signal peptides promote the transformation to the lysogenic phage [81,82,83]. Inhibition effect of repressor CI to lytic promoters shall be weakened by the binding of anti-repressor Tum with repressor CI [57]. In this study, we believed that the repressor protein and anti-repressor protein, which are both encoded by phage vB_EcoP_ZX5, are the key determinants of the lytic–lysogenic cycle.

### 3.7. E. coli YO1 Carrying the Plasmid Produced More Phage vB_EcoP_ZX5

The phage vB_EcoP_ZX5 was added to the logarithmic phase *E. coli* YO1 and plasmid-carrying *E. coli* YO1-pGEX-6p-1 for culture. After 6 h of culture, the titers of total phages in *E. coli* YO1 and *E. coli* YO1-pGEX-6p-1 were 10^9^ and 10^10^ PFU/mL, respectively, the titers of free phages in *E. coli* YO1 and *E. coli* YO1-pGEX-6p-1 were 10^7^ and 10^10^ PFU/mL, respectively, and the titers of prophages in YO1 and *E. coli* YO1-pGEX-6p-1 were 10^9^ and 10^9^ PFU/mL, respectively (Figure 7). Compared with YO1, the YO1-pGEX-6p-1 group had a higher titer of total phages and free phages. We assumed that the presence of plasmids increased the production of phages instead of changing the dominant population of phages, resulting in a change in the proportion of free phages (99%) at the later stage of culture. Assuming that each cell integrates one phage genome (the titer of bacteria is approximately 10^9^ CFU/mL), the *E. coli* YO1-pGEX-6p-1 culture still has a high titer of free phages. This hypothesis was also verified by the spontaneous induction rate of the *E. coli* YO1-pGEX-6p-1 offspring lysogenic bacteria. The titer of spontaneously induced free phages in the *E. coli* YO1-pGEX-6p-1 lysogenic bacteria was 10^7^ PFU/mL (1%), the same as that in the *E. coli* YO1 lysogenic bacteria. Studies have shown that the type and number of plasmids in bacteria affect the probability of spontaneous prophage induction [84]. In this study, the plasmid did not affect the spontaneous induction probability of phages but increased the production of phages.

### 3.8. Repressors and Anti-Repressors Affected the Lytic–Lysogenic Cycle of the Phage vB_EcoP_ZX5

The pGEX-6p-1 plasmids, which encode the GST tag, ORF4 anti-repressor, ORF12 integrase, and ORF21 repressor, were transferred into the host *E. coli* YO1 for induced expression. Western blotting was used to evaluate the expression of the target protein (Appendix A). After the control group was induced by IPTG for 6 h, the titers of total phages, free phages, and prophages in the cultures were 10^10^, 10^10^, and 10^9^ PFU/mL, respectively (Figure 8). After ORF12 integrase was induced by IPTG for 6 h, the titers of total phages, free phages, and prophages in the cultures were 10^10^, 10^8^, and 10^9^ PFU/mL, respectively (Figure 8). Integrase can specifically integrate phages into the host DNA [85,86,87]. Although the overexpression of integrase did not affect the titer of total phages, it led to a marked decrease in the titer of free phages. Strangely, the sum of the free phages and prophages titers did not match the titer of the total phages. Therefore, we speculated that the phage vB_EcoP_ZX5 might be integrated into other sites of the host genome or as a form of plasmid in cells. After ORF21 repressor was induced by IPTG for 6 h, the titers of total phages, free phages, and prophages in the cultures were 10^8^, 0, and 10^8^ PFU/mL, respectively (Figure 8). This indicates that the overexpression of the repressor protein allowed the phage vB_EcoP_ZX5 to integrate into the host genome stably, and the rapid reduction of free phages led to a significant reduction in the titer of total phages. After the ORF4 anti-repressor was induced by IPTG for 6 h, the titers of total phages, free phages, and prophages in the cultures were 10^10^, 10^10^, and 10^6^ PFU/mL, respectively (Figure 8). The results showed that the overexpression of the anti-repressor protein could be reduced instead of preventing the lysogenic cycle of the phage vB_EcoP_ZX5.

The qPCR results showed that the transcriptional levels of integrase, repressor, and anti-repressor induced by IPTG were from 500 to 1500 times higher than those in the control group (Figure 9a–c). The overexpression of phage integrase and repressors resulted in a 73% and 99.99% reduction, respectively, in phage endolysin transcription (Figure 9d). The overexpression of phage integrase and repressors resulted in a 67% and 99.97% reduction, respectively, in phage holin transcription (Figure 9e). The overexpression of the anti-repressors resulted in a 75% reduction in phage repressor transcription (Figure 9f). We believe that the ORF21 repressor of the phage vB_EcoP_ZX5 could prevent the transcription of lytic genes, whereas the ORF4 anti-repressor could slow down the transcription of lysogenic genes. In this paper, we speculated that the combination of repressor to *P_L_* operon prevents the transcription of lytic promoter *P_L_*, while the competitive combination of anti-repressor to repressor initiates the transcription of lytic promoter *P_L_*.

### 3.9. Integration of the Phage vB_EcoP_ZX5 Did Not Affect the Phenotype of E. coli YO1

Studies have shown that prophages can alter host cell phenotypes. For example, the prophage phiv142-3 in APEC strains can increase the survival rate of bacteria when they encounter acid, alkali, and oxidative stress [18]. Phage phiV10 encodes O-acetyltransferase (Oac), which can modify the O157 antigen to protect the host from invasion [58]. The prophage in *E. coli* K-12 BW25113 is beneficial for resisting osmotic, oxidative, and acid stress and promoting the growth and formation of biofilms [21,88]. In this study, the integration of vB_EcoP_ZX5 had no significant effect on the growth, the response to environmental stress and production of biofilms or the antibiotic sensitivity of *E. coli* YO1 (Figure 10).

### 3.10. Integration of the Phage vB_EcoP_ZX5 Did Not Affect the Adhesion and Invasion of Host E. coli YO1

Adhesion is one of the most important steps in bacterial infection, which is closely related to bacterial virulence. Studies have shown that some prophages could affect bacterial adhesion phenotype by coding adhesion related genes, inserting them into bacterial adhesion related genes, and regulating the transcription level of bacterial adhesion related genes [18]. HeLa cell lines were used to test the adhesion and invasion effect of phage vB_EcoP_ZX5 on host cells. Figure 11 shows that *E. coli* YO1 and *E. coli* YO1+ had similar adhesion and invasion abilities on HeLa cells. 

### 3.11. Integration of the Phage vB_EcoP_ZX5 Did Not Affect the Virulence of Host E. coli YO1

The results of serum resistance tests showed that approximately 40% of bacteria survived after treatment with mouse serum, with no significant difference between YO1 and YO1+ groups (Figure 12a). The bacteria-injected mice were euthanized two days later, and the bacterial titer in each organ was measured. The bacterial titer in each organ was approximately 10^4^ CFU/mL, with no significant difference between YO1 and YO1+ groups (Figure 12b). After the intraperitoneal injection of the mice with *E. coli* YO1 or *E. coli* YO1+ at 10^9^ CFU, the survival rate of the mice was 66% (Figure 12c). These results indicate that lysogeny of the phage vB_EcoP_ZX5 had no significant effect on the virulence of *E. coli* YO1. In contrast, some prophages have been reported to affect host virulence. For example, the phage phiv142-3 has been found to enhance the adhesion of APEC strains to DF-1 cells, resistance to serum killing, and colonization ability in chickens [18]. Similarly, the integration of the temperate phage BHP09 significantly reduced the virulence of its host *Bordetella bronchiseptica* Bb01 in mice [46]. The integration of phage vB_EcoP_ZX5 has no significant effect on the virulence of its hosts, and this may be related to the fact that the vB_EcoP_ZX5 does not encode virulence genes and drug resistance genes.

## 4. Conclusions

In this study, the temperate phage vB_EcoP_ZX5 and its host *E. coli* YO1 (O87:H52:K1) were isolated from human fecal samples. The biological characteristics, genomic characteristics, and lytic–lysogenic cycles of the phage were evaluated. The phage vB_EcoP_ZX5 was found to have a short tail and belong to the genus *Uetakevirus* and the family *Podoviridae*. The phage vB_EcoP_ZX5 integrated into the 3′-end of *guaA* of *E. coli* YO1 to form lysogenized bacteria that existed stably. The integration of vB_EcoP_ZX5 had no significant impact on the growth rate, biofilm, stress environment response, serum sensitivity, antibiotic sensitivity adherence to HeLa cells, or virulence of *E. coli* YO1. In order to identify the key proteins affecting phage lytic–lysogenic cycles, the changes in phage gene (*repressor*, *endolysin*, and *holin*) transcription level and phage titer (total phages, free phages, and prophages) caused by overexpression of ORF12 integrase, ORF21 repressor protein, or ORF4 anti-repressor protein were analyzed. The results show that the high level of ORF21 repressor completely prevented the phage from entering the lytic cycle, whereas the ORF4 anti-repressor protein could enhance the phage from entering the lytic cycle. Our findings contribute to the enrichment of the temperate phage information library, and the mystery of how repressor and anti-repressor proteins take effect on phages’ life cycles remains to be unraveled.

## Figures and Tables

**Figure 1 pathogens-11-01445-f001:**
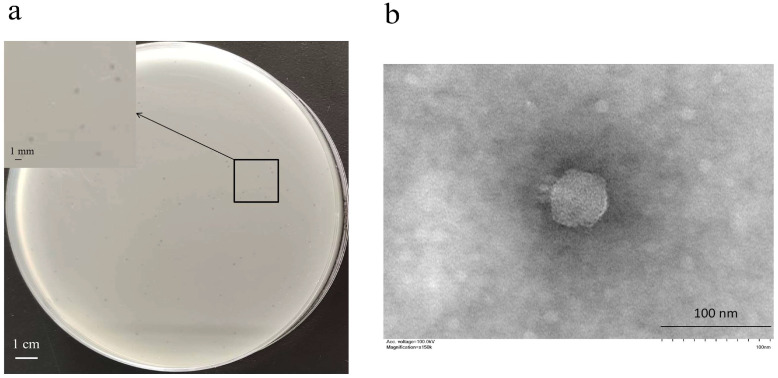
Phage morphology: (**a**) plaque and (**b**) transmission electron microscope of the phage vB_EcoP_ZX5.

**Figure 2 pathogens-11-01445-f002:**
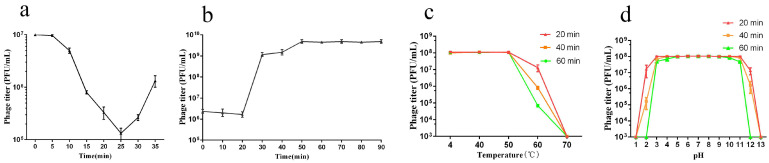
Biological characteristics of the phage vB_EcoP_ZX5. (**a**) Phage adsorption curve, (**b**) one-step growth curve, (**c**) thermal stability, and (**d**) pH stability.

**Figure 3 pathogens-11-01445-f003:**
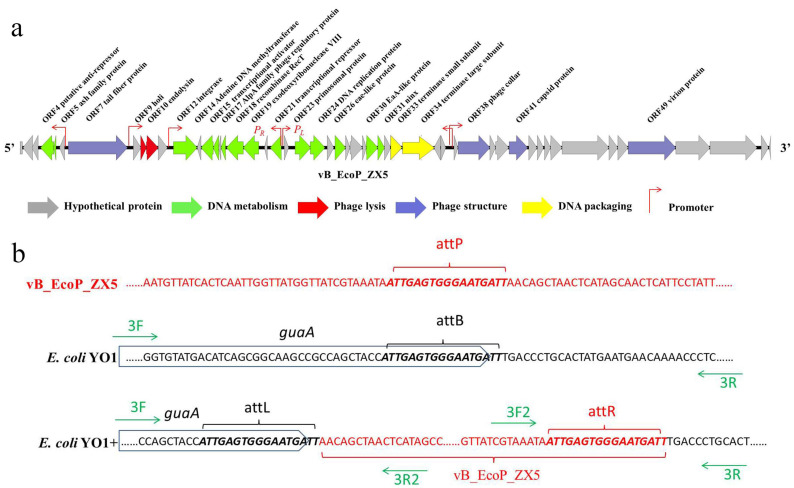
Genome characteristics of the phage vB_EcoP_ZX5. (**a**) The genomic structure of the phage vB_EcoP_ZX5. The open reading frames are divided into five classes and displayed in different colors (the clockwise arrow indicates the forward reading frame, and the counterclockwise arrow indicates the reverse reading frame). (**b**) The integration site of phage vB_EcoP_ZX5 in *E. coli* YO1+. Sequences derived from phage are marked in red, sequences derived from bacteria are shown in black, and sequences derived from the *guaA* are marked with boxes.

**Figure 4 pathogens-11-01445-f004:**
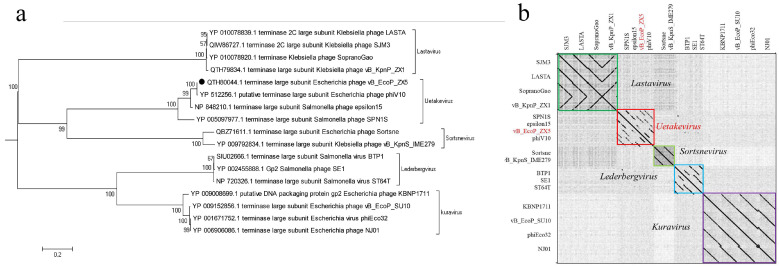
Phylogenetic analysis of the phage vB_EcoP_ZX5. (**a**) Phylogenetic tree of the terminase large subunit of the phage vB_EcoP_ZX5. The scale bar represents a phylogenetic distance of 0.2, and the numbers at the nodes represent the percent bootstrap values. (**b**) Genomes dot plot analysis of the phage vB_EcoP_ZX5. Different genome clusters are marked with different colored boxes.

**Figure 5 pathogens-11-01445-f005:**
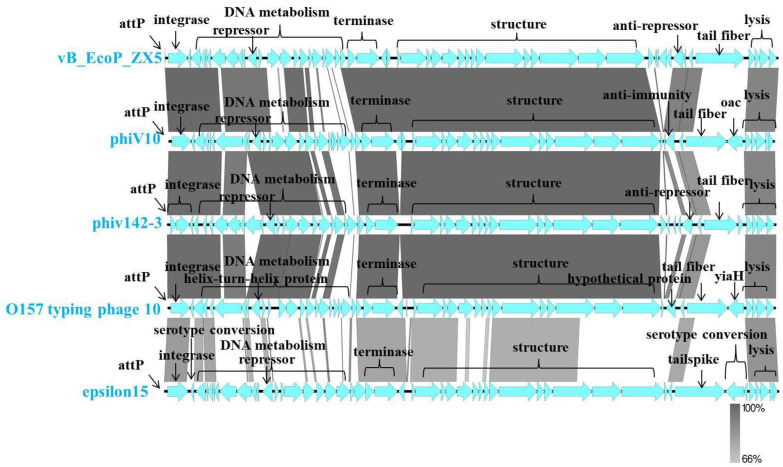
Comparative genomic analysis of the phage vB_EcoP_ZX5. The whole genome of the phage vB_EcoP_ZX5, *Escherichia coli* O157 phage type 10, *Enterobacteria* phage phiV10, *Escherichia* phage phiv142-3 and *Enterobacteria* phage epsilon15 were compared at the DNA level. The percentage of sequence similarity is displayed as green intensity.

**Figure 6 pathogens-11-01445-f006:**
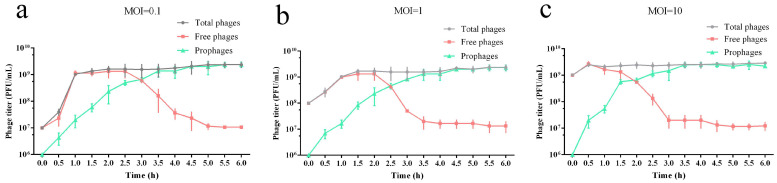
Titer of total phages, free phages and prophages under different MOI cultures: (**a**) MOI = 0.1, (**b**) MOI = 1, and (**c**) MOI = 10.

**Figure 7 pathogens-11-01445-f007:**
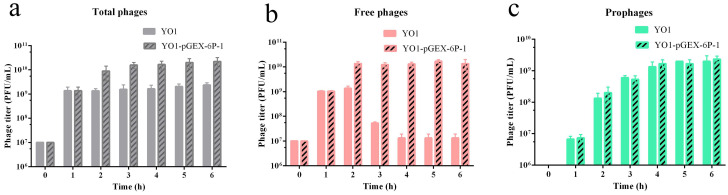
Titer of phage vB_EcoP_ZX5 cultured in *E. coli* YO1 and *E. coli* YO1-pGEX-6p-1: (**a**) total phages, (**b**) free phages, and (**c**) prophages.

**Figure 8 pathogens-11-01445-f008:**
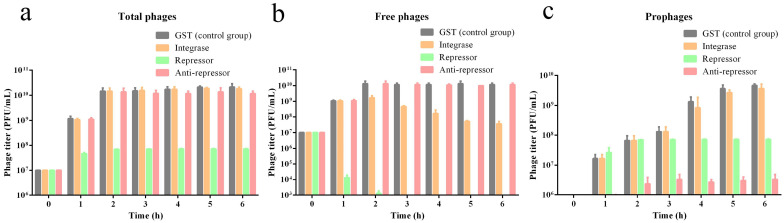
The titer of phage vB_EcoP_ZX5 cultured in gene overexpression strains: (**a**) total phages, (**b**) free phages, and (**c**) prophages.

**Figure 9 pathogens-11-01445-f009:**
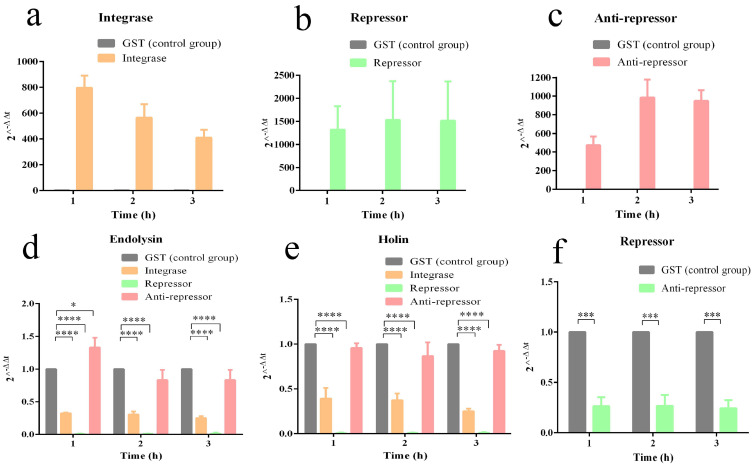
Transcription level of phage vB_EcoP_ZX5 genes. (**a**) Transcriptional level of phage integrase in integrase-overexpressing bacteria. (**b**) Transcriptional level of phage repressor in repressor-overexpressing bacteria. (**c**) Transcriptional level of phage anti-repressor in anti-repressor-overexpressing bacteria. (**d**) Transcriptional level of phage endolysin in integrase-, repressor-, and anti-repressor overexpressing bacteria. (**e**) Transcriptional level of phage holin in integrase-, repressor-, and anti-repressor overexpressing bacteria. (**f**) Transcriptional level of phage integrase in anti-repressor-overexpressing bacteria. *p* < 0.05 (*), 0.001 < *p* < 0.01 (***) and *p* < 0.001 (****).

**Figure 10 pathogens-11-01445-f010:**
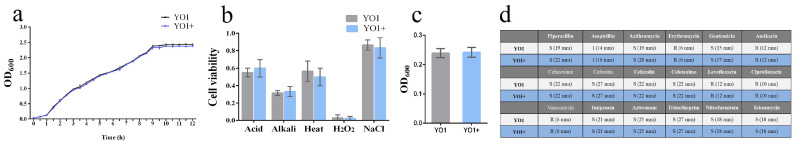
Effect of phage vB_EcoP_ZX5 integration on *E. coli* YO1 phenotype. (**a**) Bacterial growth curve, (**b**) survival rate of bacteria under external pressure, (**c**) UV absorption value of biofilm, and (**d**) bacterial antibiotic sensitivity.

**Figure 11 pathogens-11-01445-f011:**
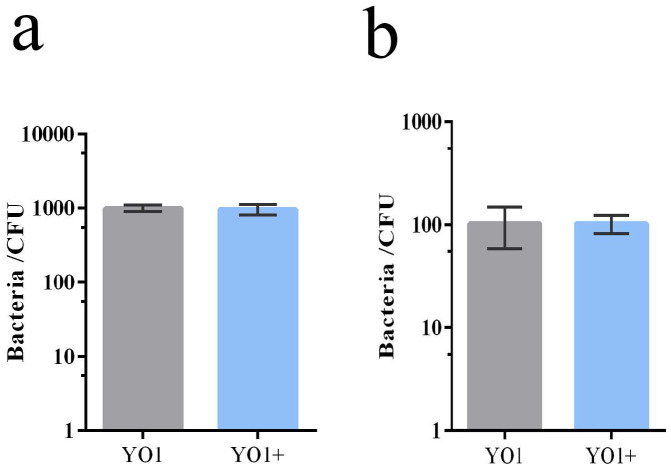
Adhesion and invasion of *E. coli* YO1 and *E. coli* YO1+. (**a**) The number of bacteria adhered to the surface of HeLa cells and (**b**) invaded into the interior of HeLa cells.

**Figure 12 pathogens-11-01445-f012:**
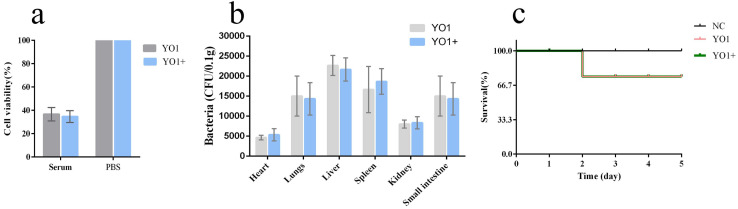
Effect of the phage vB_EcoP_ZX5 integration on virulence of host *E. coli* YO1. (**a**) Cell survival rate of bacteria after serum treatment, (**b**) bacterial colonization in mice, and (**c**) survival rate of mice after intraperitoneal injection of bacteria.

## Data Availability

The genome sequence of phage vB_EcoP_ZX5 is available from the GenBank database (accession number MW722083).

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
