# Peer review of "Characterization of a New Temperate Escherichia coli Phage vB_EcoP_ZX5 and Its Regulatory Protein"

_pathogens, 2022, doi:10.3390/pathogens11121445_

Round 1

Reviewer 1 Report

In this manuscript, Li et al. identified and characterized a new temperate E. coli phage, its regulatory protein, and how the phage affects the virulence of E. coli strain YO1. However, here are the major issues of this manuscript:

1.     In the phage stability test, the authors tested the phage thermal and pH stability at various conditions for 60 minutes. It would be informative that authors could do a time series study to show how temperature and pH affect phage stability.

2.     In line 254, the authors suggested that crystal violet was used to determine the biofilm formation. However, there is no biofilm mass in the result and discussion section.

3.     In line 301, the authors suggested that a large burst size could be caused by a high adsorption rate of phage. What is the rate in terms of the number of phages adsorbed per cell per mL per minute?

4.     For all plasmid experiments, 5x10e7 pfu/mL phage was added. However, in both figure 7 a and b, and figure 8 a and b, the phage concentration at time 0 is 1x10e7 pfu/ml.

5.     In figure 11, the authors suggest that each organ was approximately 10e4 CFU/mL, and showed the result in figure 11b. It would be nice if the authors could indicate what organs were harvested during the experiment and what CFU was in each organ.

6.     In figure 11C, the authors compared the infectivity/lethality of YO1 and YO1+. The wide type has about 66% survival rate, and organ CFU indicates that after two days, mice cleared most of the injected bacteria. It is very hard to convince how the interested phage contributes to bacterial colonization and lethality in mice.

Author Response

Reviewer 1

In this manuscript, Li et al. identified and characterized a new temperate E. coli phage, its regulatory protein, and how the phage affects the virulence of E. coli strain YO1. However, here are the major issues of this manuscript:

  1. In the phage stability test, the authors tested the phage thermal and pH stability at various conditions for 60 minutes. It would be informative that authors could do a time series study to show how temperature and pH affect phage stability.

Reply: With reference to your suggestion, we characterized the effect of time series (20, 40, 60 min) under different temperature and pH environments on phage activity. The results are modified in lines 311-314, Figure 2c and 2d of the revised manuscript. "The phage lost 87/99.93% activity after 20/60 min at 60 °C, and completely inactivated after 20 min at 70 °C (Figure 2c). Phage was slightly inactivated (30/17%) at pH=4/10 for 60 min, most inactivated (99.8/98.3%) at pH=2/11 for 40 min, and completely inactivated at pH=1/13 for 20 min (Figure 2d)."

  1. In line 254, the authors suggested that crystal violet was used to determine the biofilm formation. However, there is no biofilm mass in the result and discussion section.

Reply: The content of biofilm is measured by ultraviolet absorption value, as shown in Figure 10c. The biofilm content didn’t show a significant difference between by YO1 and YO1+.

  1. In line 301, the authors suggested that a large burst size could be caused by a high adsorption rate of phage. What is the rate in terms of the number of phages adsorbed per cell per mL per minute?

Reply: The adsorption rate is 3.9 × 10-3 PFU/cell/min/mL, which has been revised in line 306-307 of the revised manuscript.

  1. For all plasmid experiments, 5x10e7 pfu/mL phage was added. However, in both figure 7 a and b, and figure 8 a and b, the phage concentration at time 0 is 1x10e7 pfu/ml.

Reply: The typing error of "107 pfu/mL" has been revised in line 224 of the revised manuscript.

  1. In figure 11, the authors suggest that each organ was approximately 10e4 CFU/mL, and showed the result in figure 11b. It would be nice if the authors could indicate what organs were harvested during the experiment and what CFU was in each organ.

Reply: The bacterial load in the heart, lungs, liver, kidney, spleen and small intestine of mice is shown in Figure 11b of the revised manuscript.

  1. In figure 11C, the authors compared the infectivity/lethality of YO1 and YO1+. The wide type has about 66% survival rate, and organ CFU indicates that after two days, mice cleared most of the injected bacteria. It is very hard to convince how the interested phage contributes to bacterial colonization and lethality in mice.

Reply: I quite agree with you. The role of the mouse immune system in the experiment led to the elimination of most of the wild type bacteria, which made the experimental conclusion not rigorous enough. Here, I have some opinions that this experiment can preliminarily prove that the integration of phage vB_EcoP_ZX5 cannot significantly affect the pathogenicity of host E. coli YO1 to mice. If the integration of phages can increase the virulence of bacteria, it will show higher bacterial load and mortality. On the contrary, if phage integration can reduce the virulence of bacteria, it will show lower bacterial load and mortality. However, no significant changes have been observed, unless the effect of phages is negligible relative to the immune system's bacterial clearance. Combined with the previous experimental phenomenon that phage does not affect the phenotype of bacteria (growth, biofilm, environmental stress response and antibiotic sensitivity), we conclude that the integration of phage vB_EcoP_ZX5 does not affect the pathogenicity of E. coli YO1 to mice.  

Reviewer 2 Report

The manuscript by Li et al is elaborate and there are only few scientific reports that analyze an temperate phage isolated from the environment in detail. Temperature stability and pH stability of the phage was also carried out.  Comparison of the phage genome to other known phages were done. The increase in phage production with just the plasmid alone is a bit wierd. Over expression of the repressor and anti-repressor proteins showed the desired effect of control over lysis-lysogeny decision making and the use of qPCR to show changes in gene expression levels is very useful. The temperate phage did not confer any fitness benefit or a change in virulence of host. Overall the manuscript is well written and a very valuable addition to the field.

I only have couple of minor comments.

1. Can the authors elaborate a bit more on how they differentiated between prophages vs free phages using PCR. This will be very helpful for readers that are not phage experts.

2. The figures are too small and look squished. So it will be better if the figures are enlarged and given as multiple panels.

Author Response

Reviewer 2 

The manuscript by Li et al is elaborate and there are only few scientific reports that analyze an temperate phage isolated from the environment in detail. Temperature stability and pH stability of the phage was also carried out.  Comparison of the phage genome to other known phages were done. The increase in phage production with just the plasmid alone is a bit wierd. Over expression of the repressor and anti-repressor proteins showed the desired effect of control over lysis-lysogeny decision making and the use of qPCR to show changes in gene expression levels is very useful. The temperate phage did not confer any fitness benefit or a change in virulence of host. Overall the manuscript is well written and a very valuable addition to the field.

I only have couple of minor comments.

  1. Can the authors elaborate a bit more on how they differentiated between prophages vs free phages using PCR. This will be very helpful for readers that are not phage experts.

Reply: Thank you for your comments,the content has been revised in lines 191-196 of the revised manuscript. Different primers were used to distinguish the content of total phage and lysogenic phage. Free phages are separated by centrifugation,and then determined separately by PCR or double-layer method. Primers 4F/4R were used to amplify the sequence of terminase large subunit in phage vB_EcoP_ZX5, which can amplify 230bp fragments in all phage genomes including lytic and lysogenic cycle. Because primers 3F2 and 3R are derived from phage and bacteria respectively, 3F2/3R can only amplify 281bp fragments in lysogenic strains integrated with phage vB_EcoP_ZX5 in lysogenic cycle.  

  1. The figures are too small and look squished. So it will be better if the figures are enlarged and given as multiple panels.

Reply: Thank you for your comments. We have modified and enlarged the picture to make the figures on the picture clearer in the revised manuscript.

Reviewer 3 Report

The authors have extensively characterized the phage vB_EcoP_ZX5. The manuscript is well written.

Author Response

Reply: Thank you for your comments. Several language issues have been modified in the revised version in lines 31, 123, 134 and 465.

Lin 31: Earth --- earth.

Lin 123: The original sequencing data were filtered first and then assembled using New Blew 3.0. --- The original sequencing data were filtered at first and then were 123 assembled using New Blew 3.0.

Line 134: Easyfig --- Easyfig 2.2.5.

Line 465: In this study, we believed that the repressor protein and anti-repressor protein encoded by phage vB_EcoP_ZX5 are the key determinants of the lytic-lysogenic cycle. --- In this study, we believed that the repressor protein and anti-repressor protein, which are both encoded by phage vB_EcoP_ZX5, are the key determinants of the lytic-lysogenic cycle.

Round 2

Reviewer 1 Report

Thanks for your response.

Despite no difference between YO1 and YO1+ survival rates and bacterial CFU count in different organs, it is hard to justify how ZX5 is not contributing to YO1 virulence.

The IP infection model may not be the most appropriate for investigating E. coli virulence. As the authors mentioned in the results section, temperate phage could contribute to adhesion and colonization. Both in vitro and in vivo adhesion should be investigated. 

Author Response

Reply: According to the characterization of contribution of temperate phages to host virulence in the references, we further tested the influence of temperate phages on bacterial adhesion and invasion to HeLa cells. The method is revised in line 267-277, and the results are modified in lines 551-562 and Figure 11. At the cellular level, significant changes in the effect of phage vB_EcoP_ZX5 on host adhesion and invasion were not observed. Considering that phage does not affect the biofilm, environmental stress response, antibiotic sensitivity, adherence to HeLa cells and mouse mortality of the host, we still believe that phage vB_EcoP_ZX5 has no significant effect on the host YO1 virulence.

2.15. Adhesion and invasion assays

HeLa cells were used to analyze the adhesion and invasion abilities of E. coli YO1 or E. coli YO1+ [88]. In short, HeLa cells were inoculated on a 24-well plate and were cultured in a 5% (v/v) CO 2 incubator at 37 °C to form monolayer cells (105 cells per well). The logarithmic phase E. coli YO1 or E. coli YO1+ cells were resuspended in the FBS free DEME medium (107 CFU/mL), and then co-incubated with HeLa cells at 37 °C for 2h. For adhesion test, the cells were washed three times with PBS and then lysed with 0.5% (v/v) Triton X-100. The number of bacteria was calculated on LB plates. For the invasion test, after the cells were co-incubated with bacteria, ampicillin antibiotics (100 μg/mL) were added and incubated at 37 °C for 2 h to kill the bacteria on the surfaces of cell. Cells were cleaned by PBS for three times and then lysed with 0.5% (v/v) Triton X-100.

3.10 Integration of the Phage vB_EcoP_ZX5 did not affect the adhesion and invasion of host E. coli YO1

Adhesion is one of the most important steps in bacterial infection, which is closely related to bacterial virulence. Studies have shown that some prophages could affect bacterial adhesion phenotype by coding adhesion related genes, inserting them into bacterial adhesion related genes, and regulating the transcription level of bacterial adhesion related genes [18, 88]. HeLa cell lines were used to test the adhesion and invasion effect of phage vB_EcoP_ZX5 on host cells. Figure 11 shows that E. coli YO1 and E. coli YO1+ had similar adhesion and invasion abilities on HeLa cells.
